# Primary Lymphoproliferative Lung Diseases: Imaging and Multidisciplinary Approach

**DOI:** 10.3390/diagnostics13071360

**Published:** 2023-04-06

**Authors:** Luca Gozzi, Diletta Cozzi, Edoardo Cavigli, Chiara Moroni, Caterina Giannessi, Giulia Zantonelli, Olga Smorchkova, Ron Ruzga, Ginevra Danti, Elena Bertelli, Valentina Luzzi, Valeria Pasini, Vittorio Miele

**Affiliations:** 1Emergency Radiology, Careggi University Hospital, 50134 Florence, Italy; luca.gozzi@unifi.it (L.G.); edoardocavigli@yahoo.it (E.C.); chiaramoroni73@gmail.com (C.M.); caterina.giannessi@unifi.it (C.G.); giulia.zanto@gmail.com (G.Z.); olga.smorchkova84@gmail.com (O.S.); ronruzga@gmail.com (R.R.); ginevra.danti@gmail.com (G.D.); elena.bertelli3@gmail.com (E.B.); vmiele@sirm.org (V.M.); 2Italian Society of Medical and Interventional Radiology (SIRM), SIRM Foundation, 20122 Milan, Italy; 3Interventional Pneumology, Careggi University Hospital, 50134 Florence, Italy; valentinaluzzi@hotmail.com; 4Section of Anatomic Pathology, Department of Health Sciences, University of Florence, 50133 Florence, Italy; vale.pasini@gmail.com

**Keywords:** lung, lymphoma, HRCT, lung neoplasm, diagnostic imaging

## Abstract

Lymphoproliferative lung diseases are a heterogeneous group of disorders characterized by primary or secondary involvement of the lung. Primary pulmonary lymphomas are the most common type, representing 0.5–1% of all primary malignancies of the lung. The radiological presentation is often heterogeneous and non-specific: consolidations, masses, and nodules are the most common findings, followed by ground-glass opacities and interstitial involvement, more common in secondary lung lymphomas. These findings usually show a prevalent perilymphatic spread along bronchovascular bundles, without a prevalence in the upper or lower lung lobes. An ancillary sign, such as a “halo sign”, “reverse halo sign”, air bronchogram, or CT angiogram sign, may be present and can help rule out a differential diagnosis. Since a wide spectrum of pulmonary parenchymal diseases may mimic lymphoma, a correct clinical evaluation and a multidisciplinary approach are mandatory. In this sense, despite High-Resolution Computer Tomography (HRCT) representing the gold standard, a tissue sample is needed for a certain and definitive diagnosis. Cryobiopsy is a relatively new technique that permits the obtaining of a larger amount of tissue without significant artifacts, and is less invasive and more precise than surgical biopsy.

## 1. Introduction

According to the anatomical location in which they first arise, lymphoproliferative disorders of the lung are a heterogeneous group of diseases characterized by their lung manifestation in two possible ways: by a primary pulmonary involvement, or a secondary lung involvement, the latter either by hematogenous dissemination or a contiguous invasion starting from an adjacent site, such as hilar lymph nodes or thymus. Furthermore, primary disorders can also be subdivided into two groups. The first group contains reactive diseases, which comprise three main forms: nodular lymphoid hyperplasia (NLH), follicular bronchiolitis (FB), and lymphocytic interstitial pneumonia (LIP). The second group involves neoplastic disorders [1].

Primary pulmonary lymphomas (PPLs) can be defined as lung involvement by a malignant monoclonal lymphoid proliferation in a patient without extra-thoracic involvement, neither bone marrow infiltration evidence nor mediastinal mass within 3 months after initial diagnosis [2]. PPLs are uncommon, representing roughly 3–4% of extranodal lymphomas, less than 1% of all non-Hodgkin lymphomas (NHL), and 0.5–1% of all primary malignancies of the lung [3,4,5]. PPL Hodgkin’s lymphomas are the least frequent subtype, with less than 70 cases reported in the literature since 1927 [6]. The peak of incidence is between the fifth and the seventh decade, while people under 30 years old are rarely affected. Most studies report that there is not a significant difference between the two genders, despite men being slightly more affected [7,8].

The World Health Organization (WHO) classification of lung neoplasms recognizes three main types of PPL, which are, in order of decreasing frequency, B-cell primary pulmonary non-Hodgkin lymphomas, of which the most important is MALT lymphoma (mucosa-associated lymphoid tissue), diffuse large B-cell lymphoma (DLCBL), and lymphomatoid granulomatosis (LYG), which is a rare disease with approximately 500 studies described in the literature [9,10]. For whom it may concern, for secondary lymphoproliferative disorders, the incidence of lung involvement is higher in patients with Hodgkin lymphoma rather than in non-Hodgkin lymphoma, even though, in absolute terms, NHL is more frequent. The pathogenesis and the imaging features of secondary pulmonary lymphoma are variable depending on the type of primary lymphoma and its location, where the most common radiological features are lymph node enlargement, lymphangitic spread, and single or multiple nodules [11]. Most patients are often asymptomatic at diagnosis and are identified incidentally as stated by unspecific radiological findings. When present, the most common pulmonary symptoms are cough, dyspnea, and eventually chest pain with hemoptysis. General symptoms, such as fatigue, fever, and weight loss can occur in up to one quarter of patients [12,13].

## 2. Imaging Techniques and Multidisciplinary Approach

The radiological findings of PPL are mostly unspecific: chest X-ray is usually the first exam performed but is rarely diagnostic, showing undefined signs such as single or multiple opacities, pleural effusion or a mass, usually with a diameter of less than 5 cm in the case of MALT lymphoma [13]. High-Resolution Computer Tomography (HRCT) is unquestionably the imaging method of choice that permits a better evaluation of the radiological feature type (e.g., mass, ground-glass opacity, atelectasis, bronchiectasis/bronchiectasis, lymphangitic spread) and assessment of the disease extension, whether it is unilateral or bilateral and if there are enlarged lymph nodes in the main thoracic sites [14]. Even though HRCT is the radiological gold standard, all these signs are also commonly found among other diseases such as lung cancer, bacterial and fungal infections, metastases, hypersensitivity pneumonitis (HP), sarcoidosis, connective tissue diseases involving the lung and many other organs; thus, PPL could be easily mistaken for one of them [15].

As a third-level technique, magnetic resonance (MR), is less useful in the diagnosis of PPLs. Takashima et al. were the first ones to report a case of PPL using MR in 1990, describing a bulky mass with a surrounding rim of intensity in T2-weighted images, followed by a histological confirmation [16]. A similar result was reached in 1999 by Ooi et al., where they assessed the benefit of MR to distinguish if the pleura or the chest wall were infiltrated [17]. Some initial studies were performed recently by Jensen et al. and Kim et al., in which, using texture analysis in 3 Tesla MR, they were able to successfully differentiate PPL and fungal pneumonia, even though further studies are needed in the future to better characterize the MR features of PPL [18,19]. 

For all these reasons, a multidisciplinary approach with histological and immunohistochemical confirmations, alongside imaging, is critical for reaching a definitive diagnosis. Collecting an appropriate tissue sample is fundamental to successfully confirming the diagnosis and correct sampling can be reached in different ways: historically, the typical approach was via surgical lung biopsy (SLBx), even though with the progress of HRCT resolution and application of newer molecular technologies, bronchoalveolar lavage (BAL) and transbronchial lung biopsy (TBB) have progressively outclassed the surgical approach and thus become the preferred methods [20]. However, these new techniques have a limited tissue sample size, being diagnostic only in 30–50% of cases, as remarked by Liu et al., where, to collect an adequate amount of tissue for making a correct diagnosis of intravascular large B-cell lymphoma (IVLBCL), they were forced to perform a surgical lung biopsy [21,22]. Bronchoalveolar lavage (BAL) is one of the most useful diagnostic tools before the development of more precise diagnostic techniques. Studies show that a lymphocyte differential count ≥25% is suggestive of a granulomatous or lymphoproliferative lung disorder, with diagnostic confidence of around 67% in NHL and a lower assurance in Hodgkin’s disease being nearly 33%. The presence of B–lymphocyte levels greater than 10% in the fluid sample is highly suggestive of MALT lymphoma [23,24,25]. A more recent method that provides a larger lung tissue specimen for histopathological diagnosis without creating significant artifacts, while also being less invasive than surgical biopsy, is transbronchial cryobiopsy (cryo-TBB) [26].

It must be remembered that the quality of the sample obtained by cryo-TBB is strictly dependent on operator skills, with an ideal minimum dimension of 0.5 cm (in greatest dimension), whereas a tissue dimension less than 0.3 cm (in greatest dimension) does not provide any complementary diagnostic information than a traditional transbronchial biopsy. Despite the amount of tissue sample collected by cryobiopsy being significantly smaller than SLBx (where the mean single largest dimension is between 4.0 and 4.2 cm, which means that the specimen’s area obtained by SLBx is 32 times greater than a single cryobiopsy), usually, in cryo-TBB, there is enough tissue to perform a correct pathological analysis since multiple samples of a single lesion are obtained [26].

In the study proposed by Bianchi et al., cryo-TBB has been performed in 13 patients with lymphoproliferative disorders (either with primary or secondary lung involvement): this method was diagnostic in 12 cases, while a single patient had an inconclusive cryo-TBB result, which subsequently underwent a lung biopsy, which itself was inconclusive as well.

In particular, in this last study, 65 tissue specimens were collected with a mean number of samples per case of 5, a mean total area of 1161 mm^2^, and a median value of 81.01 mm^2^. Among these 65 samples, 60 were proven to be histologically adequate for a diagnosis. Even though five samples were inadequate, with findings such as the bronchial wall, fibrin, pleura, and muscle, in all 65 cases, a sufficient amount of lymphoid cells infiltration have been found to suggest a lymphoproliferative disease, proving the benefit of this new technique [27]. This is, up to now, the largest case report of lymphoproliferative lung diseases diagnosed with cryo-TBB, proving the benefit of this new technique.

Lastly, fluorodeoxyglucose positron emission tomography (18-FDG-PET) has been proven to be an important functional imaging tool, with a lung specificity and sensitivity of around 80–100% in patients suspected of lymphoproliferative lung disease, with a CT detection of a new lung nodule ≥ 1.5 cm that shows an FDG uptake greater than the mediastinal blood-pool structures. If the newly found nodule is less than 1.5 cm, the degree of uptake cannot be considered reliable in the diagnosis of lymphoproliferative disease [28,29]. 

Due to its unspecific presentation and the lack of universal guidelines that could help in the diagnostic process of lymphoproliferative lung disease, with this review, we aim to describe the main radiological parenchymal HRCT features of lymphoproliferative lung disease with their most common differential diagnosis, which could help radiologists in successfully recognizing the disease, therefore assisting in successfully recognizing the disease and hence allowing an early and effective multidisciplinary evaluation (Table 1).

## 3. Imaging Patterns

### 3.1. Consolidations

Consolidations are defined by the Fleischner Society as areas of homogeneous increased attenuation with usually poorly defined margins that obscure the underlying vessels [30]. According to some studies in which different series of PPLs were recognized and described, consolidations have been outlined as one of the main radiological patterns of primary NHL, in particular of MALT lymphoma (MALToma), which was depicted as the most predominant feature of MALToma (from 60% to 80%) in various studies [14,31,32,33,34]. These focal areas of consolidation in MALToma usually show a random distribution without topographic predominance, yet show multiple and bilateral localization mainly prevailing along the bronchovascular bundles [1,3,11,32,33,35,36].

A lobar “pneumonia-like” pattern was observed in one of five cases by McCulloch et al., in 16/56 (29%) cases of MALToma and 2/16 (12.5%) of non-MALToma in the study proposed by Chen et al. (Figure 1). 

Another feature described in this study is the bulging fissure sign, which has been found in 40% of patients with MALToma with pneumonia-like consolidations, referring to it as a displacement of the lobar fissure adjacent to the consolidation. The sign itself cannot differentiate a benign from a malignant disease; however, it can be useful in differential diagnosis, being present in *Klebsiella pneumoniae* infection, central lung cancer, or MALToma, [36,37]. On imaging, consolidations seen in CT scans may overlap with different pathologies: organizing or eosinophilic pneumonia, focal or lobar atelectasis, alveolar sarcoidosis, multifocal adenocarcinoma, carcinoma or in situ adenocarcinoma and other benign lymphoproliferative entities as LIP and NLH. The utility of a follow-up CT may show lesion stability even after an anti-inflammatory or antibiotic treatment, therefore excluding acute causes of consolidations. In these cases, if an anti-infective therapy shows no benefit, MALToma should be considered to be one of the main alternative diagnoses. In addition, another ancillary sign that could help distinguish pneumonia from lymphomatous disease is internal mammary nodal enlargement. These nodes usually are enlarged in breast cancer or lymphoma, rarely in pneumonia which commonly affects mediastinal or hilar lymph nodes. For this reason, an enlargement of internal mammary nodes in a persistent parenchymal consolidation is suggestive of lymphoma rather than pneumonia [38,39,40]. One of the most common and tricky differential diagnoses of MALToma is LIP, since the two entities may co-exist and show similar clinical, histological, and radiological features. Some radiologic findings that suggest a MALToma over LIP are consolidations and pleural effusion, whereas air-filled-cysts and “halo signs” are by far more common in LIP instead of lymphoma [1]. Furthermore, LIP is usually associated with other conditions, such as autoimmune diseases, AIDS (especially in the pediatric population), EBV infection, collagen vascular diseases, and Castleman disease [1,41,42,43,44]. A definitive diagnosis requires a biopsy, either surgical or cryo-TBB, alongside immunohistochemical analysis. Clinical information and a correct anamnesis could be helpful to carry out the differential diagnosis. 

In addition to consolidation, two common signs that can help in the diagnosis of PPLs are air bronchogram and angiogram signs. In the current literature, air bronchogram is reported to be seen in a range from 50% to 100% of patients with PPLs, being more common in MALToma-like rather than non-MALToma-like, since bronchi and bronchioles tend to be spared from tumor proliferation [14,32,36,37,45]. This could depend on the pathogenesis, since MALToma originates from bronchus-associated lymphoid tissue (BALT) infiltrating the submucosal epithelium with slow progression, leaving the bronchial lumen clear whereas non-MALToma-like tends to be more aggressive with quick and early bronchial invasion. 

Even if an air bronchogram is quite common in PPLs, it is not a specific sign of pulmonary lymphoma. It can be found in all the diseases where the airway system is spared, at least in early phases, such as in pneumonia, non-obstructive atelectasis, severe interstitial lung diseases (ILD), sarcoidosis, neoplasm (adenocarcinoma in situ in particular), hemorrhage and pulmonary edema. 

The CT angiogram sign has been defined as the presence of contrast enhancement of normal vascular structure within consolidation [33]. As for air bronchogram, the literature is united regarding the fact that angiogram is a common but non-specific sign of PPLs, being more frequent in patients with MALToma rather than in those non-MALToma-like and potentially can also exist in other entities as adenocarcinoma, post-obstructive consolidations and other lymphoproliferative diseases [3,33,36,45,46,47]. However, a CT angiogram sign could be useful in differentiating MALToma from lung adenocarcinoma in different stages, since the borders of vessels adjacent to consolidation are usually smooth and evident in lymphoma while being hazy and irregular as a result of infiltration and destruction in adenocarcinoma. 

### 3.2. Nodules and Masses

Nodules and masses are defined, according to the Fleischner Society, as having rounded or irregular opacity with margins well or poorly defined, respectively, with a maximum diameter of ≤3 cm or >3 cm [30]. The nodules and masses themselves are one of the most ambiguous and unspecific patterns in lung radiology; for this reason, it is necessary to look for additional clues that may contribute to reaching a definitive diagnosis. These two patterns, solitary or combined, are considered to be the most common radiological findings of non-MALToma-like tumors, secondary lung involvement in lymphoproliferative diseases and reactive lymphoid lesions [36,46,47,48,49] (Figure 2).

In a study conducted by Lee et al., single and multiple nodules have been established to be the most common radiological manifestation in HL, with a predominant upper lobe distribution similar to that found by Lewis et al., where a mass-like appearance was found in almost 80% of HL [7,50]. NLH, which is one of the rarest reactive lymphoid lesions, has been depicted as a solitary nodule, with an average diameter of 2 cm, which differs from MALToma and is usually bilateral with a multifocal appearance and from LIP or FB that rather tends to affect the lung in a more diffuse form [1]. Howling et al. studied 12 people with HRCT affected by FB (with a biopsy-proven diagnosis) and noticed that the main radiological findings were diffuse micronodular opacities with centrilobular (in the totality of cases) or peri-bronchial (in half of the cases) pattern of distribution [51]. In the case of FB, the disease is confined to the airways without a diffuse involvement of the interstitium, unlike LIP, even though they share a common patchy ground-glass appearance [1].

Masses and nodules are a common manifestation of non-MALToma-like PPLs, such as LYG and DLBCL. A characteristic that they both share is cavitation, which could help distinguish non-MALToma-like neoplasms from MALToma itself, even though LYG tends to have a peri-bronchial distribution with a basal predominance and a peripheral ground-glass halo, while DLBCL more frequently shows mediastinal nodes enlargements [1,4,52,53,54] (Figure 3).

Another radiological pattern that may be present in LYG is a core of ground-glass opacity surrounded by a denser consolidation with a thickness of at least 2 mm, known as the “reverse halo sign” or “atoll sign”. This finding is not specific to LYG and can be found in up to one fifth of patients with cryptogenic organizing pneumonia (COP), in some fungal infections, in SARS-CoV-2 infection, tuberculosis, sarcoidosis, granulomatosis with polyangiitis (GPA), radiation pneumonitis and many others. In immunocompromised patients, fungal infection should be considered to be the first suspect, while whether is combined with nodules inside the halo or with centrilobular opacities, the most probable differential diagnosis is with active tuberculosis [55,56,57,58].

Along with the “atoll sign”, within lymphoproliferative lung disease, another sign that can be found is the “halo sign”. This term refers to a central nodule with a peripheral ground-glass halo and, as occurs in the “reverse halo sign”, can be found in many other diseases, among these we remind: hemorrhagic metastasis, angioinvasive aspergillosis, vasculitis (if associated with cavitations can suggest GPA), infections (viral, fungal, bacterial or mycobacterial), neoplasms (primarily adenocarcinoma) [3,59,60]. A particular category of lymphoproliferative lung disorder that may mimic either LYG or angioinvasive aspergillosis, is post-transplant lymphoproliferative disorder (PTLD). These characteristic disorders usually develop within 2 years after the transplant when an EBV serum-conversion occurs in immunocompromised transplanted patients. In particular, lung involvement is more common after heart and lung transplants [61]. To distinguish LYG from PTLD, it must be considered that PTLD lacks some of the key features that are present in LYG, such as invasion or destruction of vessels or necrosis [62]. Angioinvasive aspergillosis remains the main differential diagnosis with PTLD, since the clinical scenario and the radiological findings (multiple nodules with “halo sign”) are similar for both, leading to a necessity to obtain a tissue sample to reach the correct diagnosis [63]. 

The pattern of distribution of nodules and masses in PPLs is random, with no prevalence in upper or lower lobes, yet usually, these nodules tend to show a perilymphatic spread adjacent to bronchovascular bundles or a centrilobular appearance (Figure 4). 

Perilymphatic diffusion may resemble other diseases, of which the most common are sarcoidosis, lymphangitic carcinomatosis (together with interlobular septal thickening), silicosis, work-related pneumoconiosis, amyloidosis, miliary infections and reactive lymphomatous diseases [64].

In some of these diseases, such as silicosis, sarcoidosis, and work-related pneumoconiosis, the nodules tend to be progressively coalescent and are localized more frequently in the upper/middle lobes with a highly suggestive, but not always present, eggshell calcification of enlarged mediastinal nodes (more typically seen in silicosis and pneumoconiosis whereas more evident in late-stage sarcoidosis) [65,66,67,68]. On the contrary, amyloidosis nodules tend to be localized habitually in the lower lobes with a predominance for subpleural areas, presenting a central calcification in ~50% of cases [69]. 

Lymphangitic carcinomatosis is usually related to a lymphatic spread of adenocarcinoma, where the most common is breast carcinoma followed by lung and stomach cancer [70]. Since the perilymphatic spread is undeniably a common and unspecific find, it is mandatory to collect a correct anamnesis (either familiar or work anamnesis), which has to be related to clinical and laboratory results and, if necessary, with BAL or lung biopsy, to confirm whether or not the suspicion of lymphoma.

Among lymphoproliferative lung diseases, centrilobular appearance is more common in reactive lymphomatous diseases, and the main differential diagnosis is with respiratory bronchiolitis, infections with endobronchial spread and non-fibrotic hypersensitivity pneumonia. All these conditions usually respond well to an appropriate antibiotic or glucocorticoid therapy; for this reason, it may be necessary, when under the suspicion of reactive lymphoproliferative disease, to re-evaluate the patient with HRCT after proper therapy [71].

### 3.3. Ground-Glass Opacities

The term ground-glass opacity (GGO) refers to an area or a portion of lung parenchyma with hazy increased attenuation that does not obscure the underlying bronchovascular bundles [30]. The GGO pattern alone is one of the least common radiological findings of pulmonary lymphoma, either primary or secondary, and the literature describing this radiological feature as an initial presentation is quite poor. However, some authors described patchy areas of GGO as the main presentation of PPLs in their case study, such as reported by Cozzi et al., where diffuse patchy GGOs are defined as the main radiological feature in a 30-case report, being found in 16/30 patients (53.3% of total), 13/23 of them reported to be MALToma (56.5%) and 3/7 as DLBCL (43%) [45]. In the literature, some rare cases of GGO pattern have been depicted as a primary manifestation of DLBCL. In particular, Saeki et al. were among the first to describe in the literature a case of DLBCL with patchy bilateral GGO, followed by Tokayasu et al. and other reports that described some forms of DLBCL and intravascular large B-cell lymphoma (IVLBCL) with bilateral GGO, later confirmed with TBB or surgical biopsy [72,73,74,75,76,77,78]. 

IVLBCL specifically must be considered to be a particular subtype of DLBCL, where lymphomatous cells have a growth predisposition within the vascular lumen. In the literature, both entities are considered to be diseases with a poor prognosis and the GGO appearance may be related to that, potentially being associated with a faster clinical course toward exitus. In support of this affirmation, most of the patients in the previous studies with a GGO appearance in DLBCL and IVLBCL, progressively faced worsening of respiratory function with a progression of consolidation of GGO and, in some cases, the disease spread to other sites leading to multi-organ failure and, eventually, to death [72,73,74,76]. In these cases, the GGO appearance could be explained by the hematogenous spread of the disease. Moreover, there was not a significant thickening of the bronchovascular interstitium that usually occurs when the disease spreads through the lymphatic pathway (Figure 5).

This could lead to the conclusion that the pathway of the dissemination of lymphomatous disease is for the most part via hematogenous spread instead using lymphatic pathways. As already discussed, GGO can be associated with small nodules or parenchymal consolidation in the so-called “halo sign” or “reverse halo sign”, especially in LYG and non-MALToma-like forms. GGO can be correlated as well with a superimposed thickening of the interlobular septa and intralobular lines, in the so-called “crazy-paving” pattern [30]. This appearance is typical of acute respiratory distress syndrome (ARDS), acute interstitial pneumonia (AIP), pulmonary alveolar proteinosis (PAP), and some infections (e.g., tuberculosis, COVID-19, *Pneumocystis Jirovecii pneumonia* (PCP)). At time of publication, only two case reports have been found describing a crazy-paving appearance in HRCT of MALToma, later confirmed with histological examination [79,80]. However, a “crazy-paving” appearance in HRCT has been described in some cases of secondary lung involvement, in particular by T-cell lymphomas [81,82,83]. Differential diagnosis of patchy or diffuse GGO is mandatory on admission, due to its strong correlation with the patient’s clinical conditions and bio-humoral markers of the patient. GGO itself is highly unspecific, with a broad etiology including inflammation, partial filling of air spaces, hemorrhages, edema, neoplasms with lepidic growth (adenocarcinoma in situ primarily), follicular bronchiolitis, fibrosis, poor respiration, pneumonia (eosinophilic, lipoid, viral), hypersensitive pneumonitis, and many others.

### 3.4. Interstitial Involvement

Interstitial involvement is considered a rare presentation of PPL, being more common in some reactive diseases, such as LIP, and in secondary lung lymphomatous diseases. Even though the literature reports some cases of interstitial thickening due to the infiltration of lymphomatous cells in PPLs, in particular in MALToma, this radiological pattern usually occurs in 5 to 10% of cases [14,32,36,83]. A higher percentage, around 40%, has been reported in the study by McColluch, even though the case study counted only five patients, a number too small to be considered statistically significant [37].

However, recently in the literature, some relationships between MALT and interstitial lung diseases (ILD) have been described. It was reported by Suzuki et al. that the association between chemotherapy in MALT and ILD may cause the worsening of fibrosis, leading to progressive lung failure and death [84]. This finding is aligned with research conducted by Touil et al., where a case of MALT was described in HRCT as a thickening of septal lines, nodular-reticular pattern, GGO, and honeycombing pattern in the subpleural and basal lung parenchyma. Despite chemotherapy and corticosteroid therapy, the ILD pattern progressed to ARDS, finally leading to exitus [85]. A similar conclusion was reached in the study by Mirili et al., where shown a case of extranodal marginal zone lymphoma, belonging to MALToma, showed a mosaic pattern in the lung parenchyma with subpleural bilateral diffuse thickening of the interstitium and honeycombing. Additionally, in this case, despite a complete cycle of immuno-chemotherapy, the patient died from the consequences of ARDS [86]. MALToma underlying ILD is a rare condition, but the two entities may co-exist. Moreover, MALT has to be considered one of the differential diagnoses of ILD, and the reason that obtaining a tissue sample is essential to make an accurate diagnosis, so that proper therapy may be started. Even though MALT alone has a good prognosis, the association between the two entities worsens the prognosis, usually leading to death. Khojeini et al. reported a rare case of an association between ILD and IVLBCL, confirmed only upon autopsy. It has been described in the literature that some cases of IVLBCL or DLBCL have an increased FDG-PET uptake at diagnosis, helping in considering lymphoproliferative lung disorders in the differential diagnosis when the only radiological feature was interstitial thickening with or without GGO [87,88,89]. Among the reactive spectrum of lymphoproliferative lung disorders, LIP is the most frequent entity, as reported in a series of 22 patients with histological confirmation of LIP, where thickened bronchovascular interstitium and mild interlobular septal thickening occurred, respectively, in 19/22 and 18/22 patients [90]. Regarding secondary lung involvement, as reported in the case study by Lewis et al., in a 31-patient review of subjects with either recurrent or secondary lung involvement in NHL/HL, the most common abnormality detected on CT scan in NHL was bronchovascular thickening, mimicking lymphangitis, present in 69% of cases (11/16 patients) [50]. A lower yet significant percentage, around 45%, was reported by Fujimoto et al., emphasizing that interstitial thickening is more common in secondary lung involvement in systemic NHL [91]. This radiological finding, involving both interlobular and intralobular septa in either a smooth or nodular aspect, is a feature that can occur also in many diseases. In particular, interlobular septal thickening in lymphoma can be smooth or nodular. 

Apart from lymphomatous diseases, some of the most frequent causes of smooth interstitial thickening are pulmonary edema and lymphangitic carcinomatosis, whereas some of the most common causes of nodular interstitial thickening are lymphangitic carcinomatosis, sarcoidosis, and Kaposi sarcoma. In addition to interstitial thickening, lymphangitic carcinomatosis and sarcoidosis share with PPLs, in particular, a typical perilymphatic spread and localization of lesions. For this reason, it is necessary to perform an accurate differential diagnosis by considering clinical signs, history, and other systemic symptoms in addition to the characteristic radiological distribution of lesions.

Lymphangitic carcinomatosis, among these three entities, is the one that most frequently affects the subpleural interstitium initially and, then, with a “retrograde spread”, advances progressively towards the central–perihilar lung area. On the contrary, both sarcoidosis and lymphoma usually show an opposite tendency, concerning first the central interstitium with a following “anterograde spread”, with a predominant involvement of the middle–upper lobes by sarcoidosis rather than pulmonary lymphoma [92,93,94]. 

To summarize, interstitial thickening is not a common presentation of lymphomatous disorders of the lung and, even though HRCT has become the standard exam to perform a correct diagnosis, this procedure alone may not be able to differentiate other causes of interstitial thickening. For this reason, obtaining a histological sample is mandatory.

## 4. Conclusions

HRCT is one of the most important diagnostic tools in characterizing and describing parenchymal findings suggestive of lymphoproliferative lung diseases. The most common imaging patterns are consolidations, nodules, and masses, either solitary or multiple, usually with a random distribution and a perilymphatic spread adjacent to bronchovascular bundles. Ancillary signs such as air bronchogram, CT angiogram signs, “halo signs” or “reverse halo signs” can help to rule out some differential diagnoses.

However, since a wide spectrum of pulmonary parenchymal diseases may mimic lymphomas, an early and correct clinical evaluation, alongside hematological exams and tissue samples, is needed to better contextualize imaging findings. In this sense, a multidisciplinary approach is necessary, in particular a close collaboration between hematologist, radiologist, pneumologist, and pathologist, to carry out the correct diagnosis.

## Figures and Tables

**Figure 1 diagnostics-13-01360-f001:**
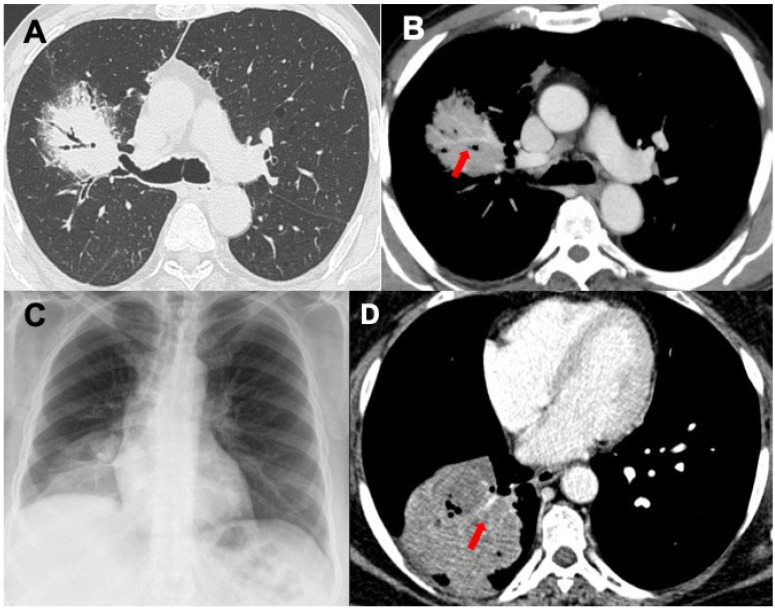
Chest CT of a primary MALT lymphoma expressed as consolidations with air bronchogram (**A**) and angiogram sign (**B**—red arrow). Images in (**C**,**D**) show a DLBCL primary lymphoma manifested as a necrotic mass: chest radiograph in (**C**) shows an opacity in the right lower lobe; the chest CT study in (**D**) confirms the presence of a mass within the angiogram (red arrow) and air bronchogram signs.

**Figure 2 diagnostics-13-01360-f002:**
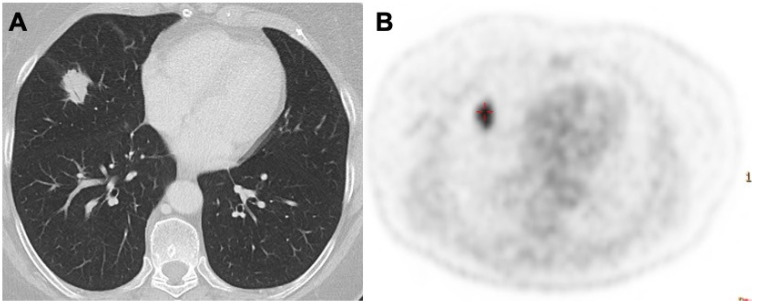
Lung nodules in the medium lobe (**A**). The PET scan in (**B**) shows a high captation of 18FDG, a high suspicion for primary lung cancer. Biopsy revealed primary MALT lymphoma of the lung.

**Figure 3 diagnostics-13-01360-f003:**
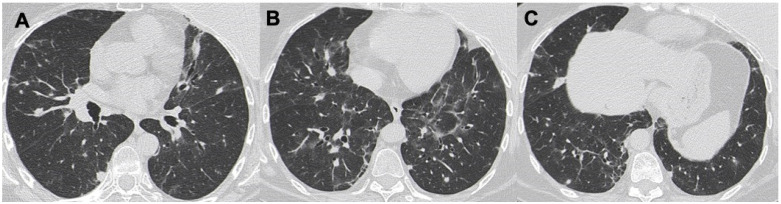
Images in (**A**–**C**) demonstrate the presence of diffuse rounded nodules in the peribronchovascular bundles; also, perilobular “atoll sign” is present, showing an “OP-like” pattern (OP: organizing pneumonia).

**Figure 4 diagnostics-13-01360-f004:**
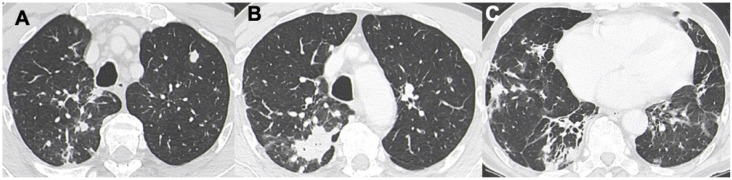
A case of DLBCL with a mixed pattern made of bilateral nodules (in **A**–**C**), interstitial involvement, and consolidations (in (**B**)).

**Figure 5 diagnostics-13-01360-f005:**
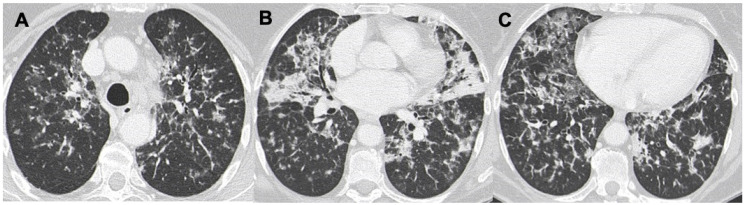
Diffuse interstitial lung involvement in MALT lymphoma. HRCT scans in (**A**–**C**) show diffuse thickening of the interlobular septa, together with peri-lobular ground-glass opacities and small confluent consolidations.

**Table 1 diagnostics-13-01360-t001:** Summary table of main radiological findings and differential diagnoses.

	Main Radiological Findings	Main Differential Diagnoses
MALT	Consolidations, bronchovascular distribution without topographic predominance, lobar-like pneumonia (rare), ±bulging fissure sign, ±angiogram sign, ±air bronchogram; less frequently nodules, masses and GGO; interstitial involvement (very rare)	Neoplasms, lobar or focal atelectasis, infections (*Klebsiella pneumoniae*), OP, LIP, NLH, sarcoidosis
DLBCL	Masses and nodules, mediastinal nodes enlargement, GGO	Neoplasms, LYG, metastasis
LIP	Air-filled cysts, “halo sign”, masses and nodules with centrilobular appearance, GGO, patchy interstitial involvement	Infections, metastasis, neoplasm, MALT, sarcoidosis, amiloidosis
NHL	Nodule (usually single) < 2 cm; less frequently mass or consolidation, air bronchogram±	Infections, neoplasm, metastasis, MALT
LYG	Masses and nodules with peribronchovascolar distribution, basal predominance, GGO, “halo sign”, “reverse halo sign”	Vasculitis, sarcoidosis, neoplasms, metastasis, PTLD, angioinvasive aspergillosi, DLBCL
PTLD	Nodules, masses or consolidations (rare), ±air bronchogram, “halo sign”, perilymphatic distribution; mediastinal nodes enlargement; interstitial thickening	Angioinvasive aspergillosis, LYG

MALT: Mucosa-associated Lymphoid Tissue; DLBCL: Diffuse Large B-cell Lymphoma; LIP: Lymphocytic Interstitial Pneumonia; NHL: Non-Hodgkin Lymphoma; LYG: Lymphomatoid Granulomatosis; PTLD: Post-Transplant Lymphoproliferative Disorder; GGO: Ground-Glass Opacities; OP: Organizing Pneumonia.

## Data Availability

No new data were created or analyzed in this study. Data sharing is not applicable to this article.

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
