# Peer review of "Primary Lymphoproliferative Lung Diseases: Imaging and Multidisciplinary Approach"

_diagnostics, 2023, doi:10.3390/diagnostics13071360_

Round 1

Reviewer 1 Report

Overall, this paper provides a comprehensive overview of lymphoproliferative disorders of the lung, radiological features, and diagnosis. The paper is well-organized and easy to follow. The authors have provided an extensive list of references, which indicates that they have thoroughly researched the topic. Overall, this paper is a valuable contribution to the literature on lymphoproliferative disorders of the lung. With a few minor revisions, it could be an even stronger resource for clinicians and researchers.

Author Response

Thank you for the revision, we have improved the English language.

Reviewer 2 Report

Well written review of a rare topic. Main focus is on HRCT findings. The article would increase interest, if the described findings are put together in an algorithm or a tabel to have the information in condensed way. In addition, diagnosis of lymphomatous diseases (especially the rare ones) is based on tissue. In lymphoma diagnosis, size of biopsy matters much more than in solid cancer. Therefore more should be said about the method to gain sufficient material in the different entities (transthoracal biopsy, transbronchial biopsy, BAL, surgery). My suggestion would be a table:

Entity -> typcial radiological findings -> method to get material -> pathology

or if you start from the radiologic finding:

typical finding -> possible entitiy and non lymphoid DD -> method to get material -> pathology

some typos: page 2: 500-600 studies? reports; page 5: citation [3,33,36,465-47]?; page 7 laboratoristic?; figure 5 interstial?; page 9 tipically? 

Author Response

Thank you for the revision, we have improved the English language and some spelling errors. Moreover, we added a table (as suggested) and a paragraph about interventional methods (BAL, biopsy, criobiopsy, surgery).